# Mortality Prediction Analysis among COVID-19 Inpatients Using Clinical Variables and Deep Learning Chest Radiography Imaging Features

Xuan V. Nguyen [1,*], Engin Dikici [1,*], Sema Candemir [1,†], Robyn L. Ball [2] and Luciano M. Prevedello [1]

[1] Laboratory for Augmented Intelligence in Imaging, Department of Radiology, The Ohio State University College of Medicine, Columbus, OH 43210, USA; candemirsema@gmail.com (S.C.); luciano.prevedello@osumc.edu (L.M.P.)
[2] The Jackson Laboratory, Bar Harbor, ME 04609, USA; robyn.ball@jax.org
[*] Correspondence: xuan.nguyen@osumc.edu (X.V.N.); engin.dikici@osumc.edu (E.D.)
[†] Current Affiliation: Department of Computer Engineering, Eskişehir Technical University, Eskişehir 26555, Turkey.

**Abstract:** The emergence of the COVID-19 pandemic over a relatively brief interval illustrates the need for rapid data-driven approaches to facilitate clinical decision making. We examined a machine learning process to predict inpatient mortality among COVID-19 patients using clinical and chest radiographic data. Modeling was performed with a de-identified dataset of encounters prior to widespread vaccine availability. Non-imaging predictors included demographics, pre-admission clinical history, and past medical history variables. Imaging features were extracted from chest radiographs by applying a deep convolutional neural network with transfer learning. A multi-layer perceptron combining 64 deep learning features from chest radiographs with 98 patient clinical features was trained to predict mortality. The Local Interpretable Model-Agnostic Explanations (LIME) method was used to explain model predictions. Non-imaging data alone predicted mortality with an ROC-AUC of 0.87 ± 0.03 (mean ± SD), while the addition of imaging data improved prediction slightly (ROC-AUC: 0.91 ± 0.02). The application of LIME to the combined imaging and clinical model found HbA1c values to contribute the most to model prediction (17.1 ± 1.7%), while imaging contributed 8.8 ± 2.8%. Age, gender, and BMI contributed 8.7%, 8.2%, and 7.1%, respectively. Our findings demonstrate a viable explainable AI approach to quantify the contributions of imaging and clinical data to COVID mortality predictions.

**Keywords:** COVID; computer-aided diagnosis/prognosis; chest radiographs; multi-modal analysis; transfer learning; machine learning; explainable AI

## 1. Introduction

Following the identification of a novel coronavirus (SARS-CoV-2) in late 2019 [1,2], there have been over 500 million diagnoses of the resultant respiratory disease (COVID-19) and over 6 million resultant deaths [3]. In severe cases of COVID-related pneumonia, diffuse alveolar damage is identifiable from post mortem pathology assessments [4], and COVID-related lung inflammation in affected patients can be detected with relatively high sensitivity during chest CT imaging [5]. Given the status of this disease as an ongoing pandemic and potential risks of future variants or new pandemics, there is a growing need for the development of data-intensive methods to assist clinical prognosis, diagnosis, or other decision making based on imaging or other medical data.

Artificial intelligence (AI)-based approaches, such as those employing machine learning methods, have been used in a variety of settings in medical image analysis to extract clinically useful information and have been proposed to assist the pandemic response [6]. Data-driven models can process large amounts of multimodal data and learn or identify patterns

embedded within large datasets. Many existing AI-based radiology studies on COVID-19 have addressed the task of detecting or diagnosing COVID-19 [7–10]. As summarized in [11], AI-based detection of chest radiographic or CT findings has been extensively studied in recent years, but there is a relative paucity of AI applications available to make other actionable predictions beyond diagnosis or detection, such as risk assessment and prognosis.

In the current study, we developed and tested a machine learning model that takes a combination of demographic, medical history, and early laboratory variables and chest radiography images as inputs and predicts mortality during inpatient encounters with COVID-positive patients. The aims of our study were twofold: (1) to develop a computational analytic framework for quantifying and explaining the contributions of imaging and non-imaging data in predicting a specified outcome variable (encounter-level mortality) and (2) to analyze and interpret the mortality prediction model obtained via training and testing on a large COVID-19 patient dataset acquired early during the pandemic prior to widespread vaccine availability. We sought to answer the following questions for COVID-positive inpatients: (1) How well do demographic and other non-imaging clinical variables predict mortality during an inpatient encounter? (2) Does a prediction model applied to a combination of chest radiography and non-imaging data improve prediction over the use of clinical data alone?

## 2. Materials and Methods

For predicting the mortality outcome, we introduced a framework processing both clinical (i.e., demographics, pre-admission clinical history, and past medical history) and imaging (i.e., chest radiography) data. It consisted of two neural networks: (1) a transfer-learned [12] VGG-16 network [13] to produce imaging features and (2) a custom multi-layer perceptron (MLP) to process clinical and imaging features to produce the mortality predictions (see Figure 1). In this section, (1) data information, (2) the prediction framework architecture, and (3) the methodology for deriving prediction explanations are provided.

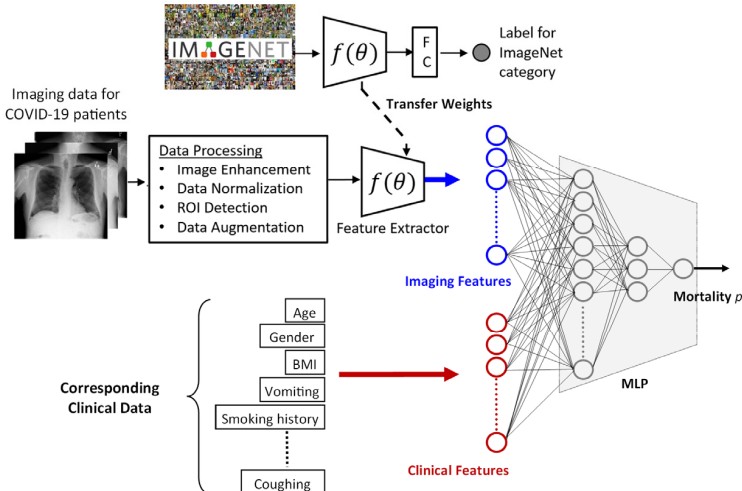

**Figure 1.** The overview of the mortality prediction framework. The imaging features are acquired using transfer-learned VGG-16 (*f*) and are subsequently processed in tandem with the clinical data features via an MLP to produce the patient's mortality probability.

### 2.1. Data Source

This study used a public de-identified dataset (COVID-19-NY-SB) containing both imaging and non-imaging clinical data [14,15]. This dataset consisted of clinical and imaging data extracted from encounters with confirmed PCR-positive COVID-19 patients. For patients with multiple encounters containing or within 7 days of a positive COVID PCR test, an encounter selection algorithm was used that prioritized the most severe encounter. The dataset contained 130 variables encoding demographic descriptors, encounter type, presenting symptoms, medical history and comorbidities, medication history, laboratory and vitals data, and outcomes, including the primary outcome of interest for this study (last status of

death vs. discharge for the encounter). Our analysis examined only the subset of inpatient encounters in the dataset for which chest radiographs were available. The acquisition parameters for the chest radiographs included in our study were as follows: kilovoltage peak: median 90 kVp (inter-quartile range (IQR): 90–90); exposure: median 2.2 mAs (IQR: 2.2–3.2); and field of view: median 353 × 421 mm (IQR: 350 × 421–354 × 424). Exemption from IRB review was formally obtained from our institutional IRB based on our study's secondary use of de-identified data.

### 2.2. Prediction Framework

#### 2.2.1. Imaging Features

The framework processed chest X-ray (CXR) images for a given patient to produce imaging features. Briefly, (1) the bounding box containing the lung fields was extracted from a given CXR; (2) the extracted region of interest was then rescaled and processed by a classification network (i.e., VGG-16), which was transfer-trained for the mortality prediction task; and (3) the latent space representation extracted from this network was utilized to produce the imaging features. These stages are further detailed in the following paragraphs.

The bounding box extraction used a segmentation-based approach in which a U-Net [16] was trained using (1) the Japanese Society of Radiological Technology's (JSRT) CXR dataset [17] and (2) their corresponding segmentation masks that were publicly available [18]. The motivation for using the U-Net formulation for the task was its proven effectiveness in medical image segmentation, which has led to its utilization in various studies over the recent years [19,20]. The JSRT dataset contained 247 CXRs: (1) 154 CXRs with lung nodules and (2) 93 CXRs without lung nodules. Each CXR image had 2048 × 2048 pixels, spatial resolution of 0.175 mm/pixel, and a 12-bit grayscale color depth. An example set of segmentation results of our U-Net is shown in Figure 2.

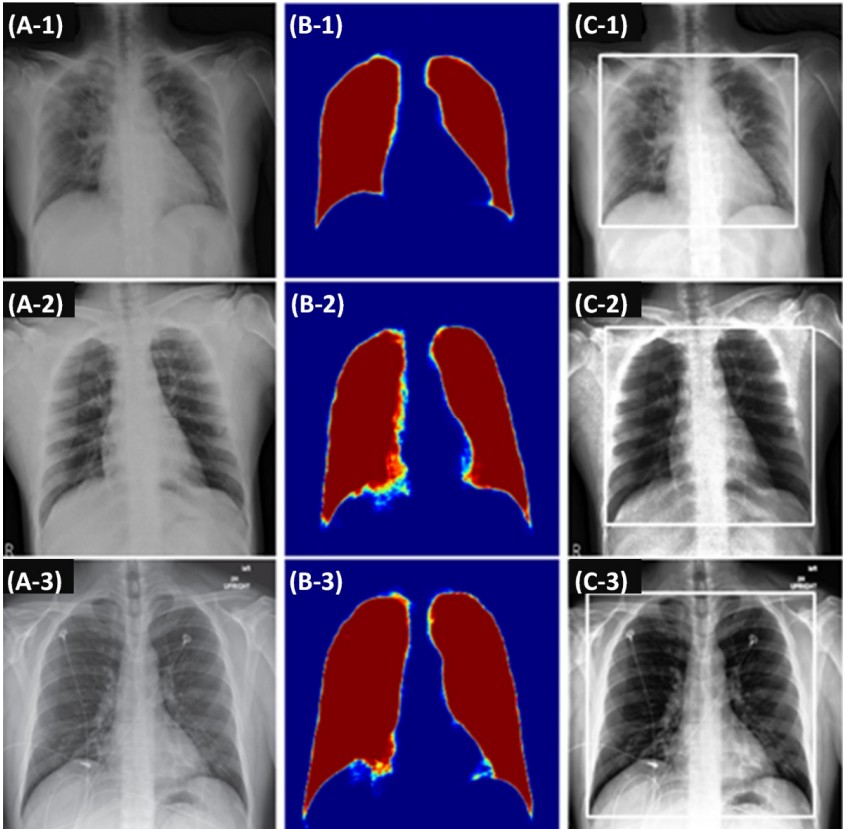

**Figure 2.** Lung region detection component applied on example CXRs. First column (**A-1** to **A-3**): example chest radiographs; middle column (**B-1** to **B-3**): predicted lung probability maps; third column (**C-1** to **C-3**): lung regions in bounding boxes suitable to process in a convolutional neural network.

The imaging features were extracted from the bounding box contents using a transfer learning approach. For the task, the extracted regions were rescaled to fixed dimensions (width: 600×, height: 800 pixels) with three identical channels (i.e., grayscale value was copied to RGB channels). Next, the VGG-16 network, pre-trained with ImageNet [21], was fine-tuned using the rescaled lung regions and their corresponding target outputs. For this analysis, the target output was a binary value representing a patient's last recorded status as 0 (discharged) or 1 (deceased). Accordingly, (1) the network was updated to have a single-node output, and (2) the dense layer before the output node was resized to the desired imaging feature vector length (i.e., latent size) before the fine-tuning. The network's latent-sized intermediate output was later utilized as the imaging feature vector.

### 2.2.2. Clinical Features

Of the available variables in the public dataset, we selected those encoding demographic descriptors, past medical history and comorbidities, pre-admission medication and symptom history, and selected laboratory or vitals data that were clearly specified in the dataset documentation as having been obtained at or prior to admission or were deemed unlikely to change over the course of the encounter (such as BMI). Most of the laboratory and vitals data within the dataset were not included in our analysis because (1) lack of precise information on the timing of acquisition of most of the laboratory and vitals data in the dataset would compromise our intended goal of restricting clinical predictors to early clinical data available at or prior to admission and (2) the large number of entries with missing values for laboratory data raises the risk of bias related to inclusion of certain laboratory data that are more likely to be obtained in higher-severity encounters. Table 1 describes the clinical data used in this study, including the value types. The information was featurized by (1) one-hot encoding of the categorical fields and (2) normalizing each categorical encoding and value field (e.g., integer, float, etc.) to the [0,1] range. As missing data are commonly encountered in clinical or epidemiological datasets, and it is often important for decision systems to be able to operate in the context of missing clinical data, we opted to designate missingness status as a separate level of each categorical variable.

**Table 1.** Clinical data variables used for prediction.

| | Field Type | Variable Description |
|---|---|---|
| Age | Categorical: [18, 59], (59, 74], (74, 90] | Age intervals (in years) at admission; truncated for patients > 90 years of age. |
| Gender | Categorical: Male, Female, Unknown/Missing | Documented gender in the electronic health record; dropped in select cases for de-identification. |
| Kidney replacement therapy | Categorical: Yes, No, Unknown/Missing | Documented renal replacement therapy. |
| Kidney transplant | Categorical: Yes, No, Unknown/Missing | History of kidney transplant. |
| Hypertension | Categorical: Yes, No, Unknown/Missing | Documented ICD-10 code for hypertension, taking anti-hypertensive medications, and/or documented blood pressure > 140/90. |
| Diabetes mellitus | Categorical: Yes, No, Unknown/Missing | Documented ICD-10 code for diabetes type 1 or 2 or taking insulin or oral medications for diabetes. |
| Coronary artery disease | Categorical: Yes, No, Unknown/Missing | Documented ICD-10 code for coronary artery disease, history of stent placement, or existing catheter report documenting disease. |
| Heart failure | Categorical: HFpEF, HFrEF, No, Unknown/Missing | For HFrEF, documented ICD-10 code for HFrEF or echocardiogram documenting reduced ejection fraction (reduced EF is <40%, 40% or higher is preserved EF). For HFpEF, documented ICD-10 code for HFpEF or echocardiogram documenting diastolic dysfunction. |
| Chronic kidney disease | Categorical: Yes, No, Unknown/Missing | Documented ICD-10 code for chronic kidney disease or reduced GFR on lab work. |
| Malignancy | Categorical: Yes, No, Unknown/Missing | Documented ICD-10 code for malignancies or receiving treatment for active malignancy. |
| COPD | Categorical: Yes, No, Unknown/Missing | Documented ICD-10 code for chronic obstructive pulmonary disease or pulmonary function tests documenting obstructive defect along with positive smoking history. |
| Other lung disease | Categorical: Yes, No, Unknown/Missing | Documented ICD-10 code for other lung diseases including asthma, interstitial lung disease, pulmonary hypertension, chronic pulmonary embolism, or lung resection. |
| Smoking status | Categorical: Current, Former, Never, Unknown/Missing | Patient's smoking status as either Current, Former, Never Smoker, or Unknown. This referred only to cigarettes and cigars. E-cigarettes and marijuana were not counted. |
| ACE inhibitor use | Categorical: Yes, No, Unknown/Missing | Admission medication reconciliation documenting use of an ACE inhibitor as a home medication. |
| Angiotensin receptor blocker use | Categorical: Yes, No, Unknown/Missing | Admission medication reconciliation documenting use of an angiotensin receptor blocker as a home medication. |
| Antibiotic use | Categorical: Yes, No, Unknown/Missing | On an antibiotic prior to presentation. |
| NSAID use | Categorical: Yes, No, Unknown/Missing | Admission medication reconciliation documenting use of a non-steroidal anti-inflammatory drug as a home medication. |
| Cough | Categorical: Yes, No, Unknown/Missing | Reported cough on admission. |
| Dyspnea on admission | Categorical: Yes, No, Unknown/Missing | Reported shortness of breath on admission. |
| Nausea | Categorical: Yes, No, Unknown/Missing | Reported nausea on admission. |
| Vomiting | Categorical: Yes, No, Unknown/Missing | Reported vomiting on admission. |
| Diarrhea | Categorical: Yes, No, Unknown/Missing | Reported diarrhea on admission. |

**Table 1.** *Cont.*

| | Field Type | Variable Description |
|---|---|---|
| Abdominal pain | Categorical: Yes, No, Unknown/Missing | Reported abdominal pain on admission. |
| Subjective fever | Categorical: Yes, No, Unknown/Missing | Subjective or objective fever at home. Fever in ED was not counted. |
| Days symptomatic | Integer value or Unknown/Missing | The number of days prior to presentation that symptoms began. |
| BMI | Categorical: <30, [30, 35], >35, Unknown/Missing | Body mass index (kg/m$^2$) |
| HbA1c | Categorical: <6.5, [6.5, 7.9], >7.9, Unknown/Missing | Hemoglobin A1c (%) |
| Temperature over 38 C | Categorical: Yes, No, Unknown/Missing | Temperature at time of admission over 38 degrees centigrade. |

Abbreviations: HFpEF: heart failure with preserved ejection fraction; HFrEF: heart failure with reduced ejection fraction.

2.2.3. Processing Imaging and Clinical Features

The extracted imaging and clinical features were concatenated and processed by an MLP consisting of two hidden dense layers, consisting of 32 and 4 units with rectified linear activations and each followed by a dropout layer (with a dropout rate of 0.1). The dense layers were initialized using the Glorot uniform initializer [22]. The sigmoid-type network output predicted the probability of patient in-hospital mortality.

*2.3. Deriving Prediction Explanations*

The explanation for a machine learning (ML) system output may be summarized as forming a human-understandable (e.g., visual or textual) relationship between the system's inputs and the corresponding output. The topic has gained significant importance in recent years [23]. Highly parametric architectures of state-of-the-art ML approaches (e.g., deep-learning-based prediction models) make it difficult to trace the underlying logic of the models' decisions. In this study, we utilized Local Interpretable Model-Agnostic Explanations (LIME) [24] to derive the explanations of the framework's output, giving the relationship between the clinical and imaging features and the mortality probability.

LIME explains each input/output pair by approximating the model's behavior locally around the samples. Briefly, the method (1) generates a random group of sample inputs around an original training input and (2) derives the input features' contributions using a regression between these random samples' proximity to the original sample and their corresponding outputs. Accordingly, LIME can produce a normalized contribution vector ($c$) for each sample:

$$c_s = (f_1, f_2, \cdots f_n) \tag{1}$$

$$with \sum_i^N f_i = 1 \tag{2}$$

where $s$ is the sample identifier and $n$ is the feature count. We computed the global explanations, summarizing the contributions of features, by finding the mean and standard deviation of normalized contribution vectors over all samples (i.e., $\bar{c}$ and $c_{std}$, both vectors of length $n$). The process allowed us to visualize and understand the model's attention distribution over the features during the prediction process.

*2.4. Statistical Analysis*

Comparison of the distribution of the outcome variable across 10 categorical patient variables was performed using the chi-square test of independence, with statistical significance assessed based on an unadjusted $p$ value less than an adjusted $\alpha$ of 0.005 (Bonferroni adjustment for 10 multiple comparisons). Performance of predictive models was quantified using areas under the ROC curve (ROC-AUC). Data for continuous variables are reported as means $\pm$ standard deviations where appropriate. Differences in variance between models was assessed using the Brown–Forsythe test. Determination of statistical significance was made based on $\alpha$ of 0.05.

**3. Results**

*Patient Population*

The data used for our analysis consisted of 5056 chest radiographs across 841 COVID-positive inpatients, balanced for the mortality outcome variable, such that (1) 2486 radiographs (49%) were of patients who died during the hospital encounter and (2) 2570 radiographs (51%) were of discharged patients. The mean length of stay was 34 days. The demographic and clinical characteristics of the patient population are shown in Table 2.

**Table 2.** Characteristics of patients included in the analysis.

| | | | In-Hospital Mortality | | | | | |
|---|---|---|---|---|---|---|---|---|
| | **All** | **Column %** | **Yes** | **Column %** | **No** | **Column %** | ***p* Value** [1] | |
| Total patients | 841 | | 180 | | 661 | | | |
| Gender | | | | | | | <0.0001 | * |
| Male | 489 | 58% | 97 | 54% | 392 | 59% | | |
| Female | 321 | 38% | 54 | 30% | 267 | 40% | | |
| Not recorded | 31 | 4% | 29 | 16% | 2 | 0% | | |
| Age | | | | | | | <0.0001 | * |
| 18–59 | 380 | 45% | 29 | 16% | 351 | 53% | | |
| 60–74 | 252 | 30% | 65 | 36% | 187 | 28% | | |
| >75 | 209 | 25% | 86 | 48% | 123 | 19% | | |
| Comorbidities | | | | | | | | |
| Hypertension | | | | | | | <0.0001 | * |
| Yes | 377 | 45% | 108 | 60% | 269 | 41% | | |
| No | 339 | 40% | 43 | 24% | 296 | 45% | | |
| Not recorded | 125 | 15% | 29 | 16% | 96 | 15% | | |
| Diabetes mellitus | | | | | | | 0.16 | NS |
| Yes | 215 | 26% | 55 | 31% | 160 | 24% | | |
| No | 503 | 60% | 97 | 54% | 406 | 61% | | |
| Not recorded | 123 | 15% | 28 | 16% | 95 | 14% | | |
| Coronary artery disease | | | | | | | <0.0001 | * |
| Yes | 131 | 16% | 49 | 27% | 82 | 12% | | |
| No | 582 | 69% | 99 | 55% | 483 | 73% | | |
| Not recorded | 128 | 15% | 32 | 18% | 96 | 15% | | |
| Heart failure | | | | | | | <0.0001 | * |
| Yes | 62 | 7% | 33 | 18% | 29 | 4% | | |
| No | 647 | 77% | 114 | 63% | 533 | 81% | | |
| Not recorded | 132 | 16% | 33 | 18% | 99 | 15% | | |
| Chronic kidney disease | | | | | | | 0.020 | NS |
| Yes | 69 | 8% | 23 | 13% | 46 | 7% | | |
| No | 645 | 77% | 126 | 70% | 519 | 79% | | |
| Not recorded | 127 | 15% | 31 | 17% | 96 | 15% | | |
| Malignancy | | | | | | | 0.034 | NS |
| Yes | 69 | 8% | 23 | 13% | 46 | 7% | | |
| No | 638 | 76% | 127 | 71% | 511 | 77% | | |
| Not recorded | 134 | 16% | 30 | 17% | 104 | 16% | | |
| Chronic obstructive pulmonary disease | | | | | | | 0.0053 | NS |
| Yes | 56 | 7% | 21 | 12% | 35 | 5% | | |
| No | 660 | 78% | 129 | 72% | 531 | 80% | | |
| Not recorded | 125 | 15% | 30 | 17% | 95 | 14% | | |
| Other lung disease | | | | | | | 0.77 | NS |
| Yes | 109 | 13% | 24 | 13% | 85 | 13% | | |
| No | 605 | 72% | 126 | 70% | 479 | 72% | | |
| Not recorded | 127 | 15% | 30 | 17% | 97 | 15% | | |

[1] $p$ values based on chi-square test of independence, with df = 2, uncorrected for multiple comparisons. The * denotes comparisons that are significant based on a Bonferroni-adjusted $\alpha$ of 0.005.

Non-imaging data alone, based on the variables shown in Table 1, predicted mortality with an ROC-AUC of $0.87 \pm 0.03$ (Figure 3a). At a default probability threshold of 0.5, sensitivity was $0.63 \pm 0.08$ and specificity was $0.87 \pm 0.04$. The addition of deep learning features extracted from chest radiographs to the model improved prediction slightly (ROC-AUC $0.91 \pm 0.02$) (Figure 3b). This corresponded to a sensitivity of $0.74 \pm 0.10$ and a specificity of $0.87 \pm 0.04$ at the default probability threshold. In addition, performance was more consistent; with the Brown–Forsythe test, the variance in the errors of the model that included both clinical and imaging data (0.145) was significantly lower ($p < 0.001$; F = 42.5, df = 1, 10,052.9) than the variance in the errors in the model using clinical data alone (0.189).

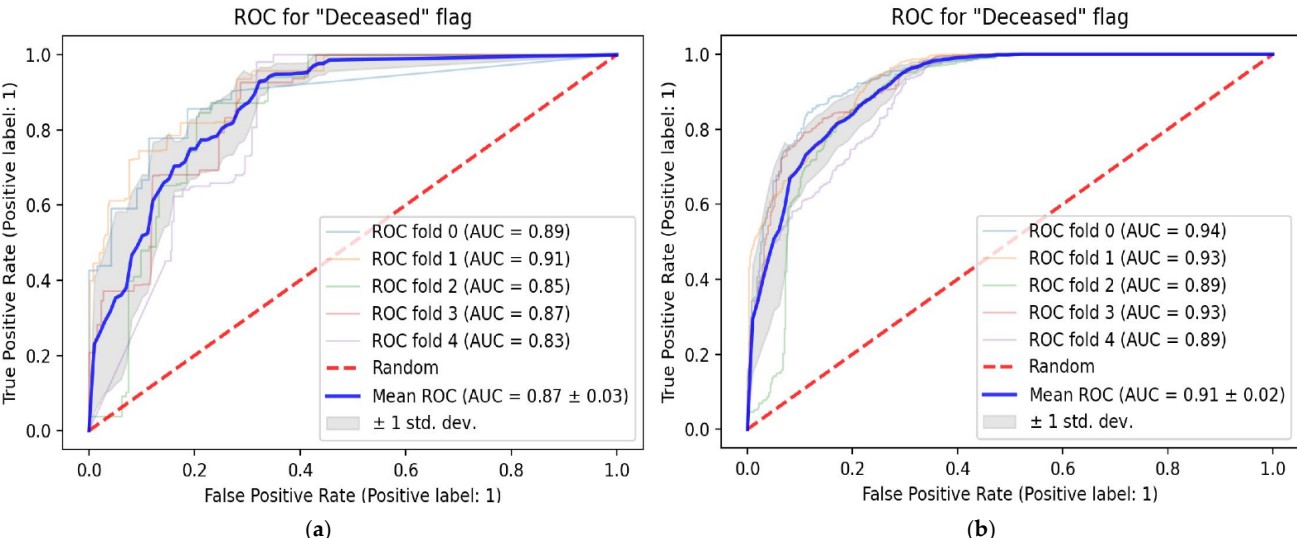

(**a**)        (**b**)

**Figure 3.** ROC curves showing the performance of prediction models. (**a**) Performance of the model using demographic and non-imaging clinical variables only; (**b**) performance of the model using imaging data and demographic and other clinical variables.

Following the application of LIME to the combined imaging and clinical model, we quantified the relative contributions of variables to the prediction of in-hospital mortality. HbA1c values contributed the most to model prediction (mean contribution of $17.1 \pm 1.7\%$), while imaging contributed $8.8 \pm 2.8\%$. Age, gender, and BMI contributed 8.7%, 8.2%, and 7.1%, respectively (Figure 4). The top five variables contributed approximately 50% of the prediction.

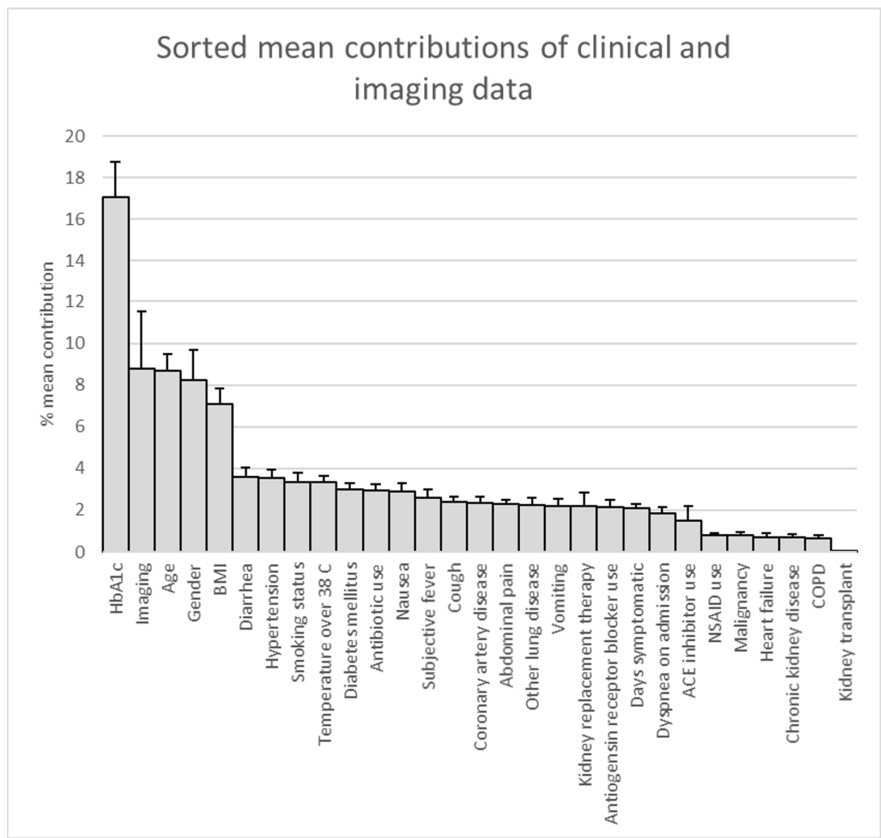

**Figure 4.** Pareto chart for the LIME analysis results for the predictive model using imaging and demographic and other clinical variables. Predictors are shown in order of descending mean contributions.

The values of each variable contributing to higher or lower mortality are shown in Table 3 for clinical variables contributing more than 5%. For instance, mortality was increased for HbA1c above 6.6, male gender, age above 74 years, and BMI over 30.

**Table 3.** Effects of specific values of the most predictive clinical variables on increasing or decreasing mortality.

|  | Reduced Mortality | Increased Mortality |
|---|---|---|
| Age | 18–59 | 74–90 |
| Gender | Female | Male |
| BMI | Below 30 | Over 30 |
| HbA1c | <6.6 | >6.6 |

In addition, we assessed the performance of the model utilizing combined imaging and non-imaging clinical data after partitioning the population into two groups based on whether patients were admitted to the ICU. In our analyzed inpatient population, for which a balanced mix of fatal and nonfatal cases was selected, 79% of subjects had an ICU admission. In-hospital mortality was 44% for the ICU subgroup and 12% for the non-ICU subgroup. The generated ROC curves are shown for the ICU subgroup (Figure 5a) and the non-ICU subgroup (Figure 5b). The model predicted mortality with an ROC-AUC of $0.93 \pm 0.03$ for the ICU subgroup and $0.83 \pm 0.05$ for the non-ICU subgroup.

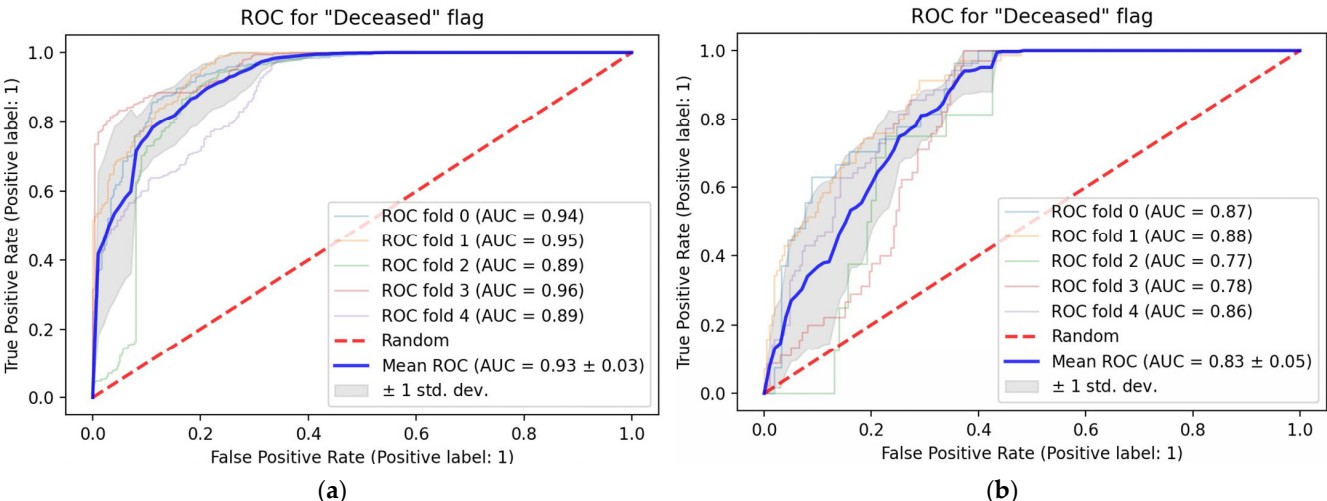

**Figure 5.** ROC curves showing performance of the prediction model utilizing combined imaging and non-imaging clinical data following partitioning based on ICU admission status. (**a**) Test performance with the ICU subgroup; (**b**) test performance with the non-ICU subgroup.

## 4. Discussion

The current study showed that our best model, which used a combination of imaging and non-imaging clinical data, predicted in-hospital mortality in COVID-19 patients with an ROC-AUC of $0.91 \pm 0.02$. The largest contributions to prediction, in descending order based on our LIME analysis, were HbA1c, imaging features, gender, age, and BMI. The performance of this combined model was greater and variance across folds appeared lower than when using non-imaging clinical data alone. In addition, test performance of our best model was better for the subgroup admitted to the ICU than for the smaller non-ICU subgroup, likely because 79% of the patients were admitted to the ICU. While a chest radiograph may contain clues, such as the presence of lines and tubes, that could assist in the determination of ICU status, the high performance within the ICU subgroup suggested that the model was not relying solely on clues of ICU status and likely made use of additional imaging findings to discriminate between fatal and nonfatal cases within this high-risk group.

While several studies have predicted COVID diagnosis or prognosis based on either clinical data separately or imaging data separately, few have quantified the relative contribution of imaging data vs. other clinical data within a combined multi-modal, multi-data prediction strategy. Our results showed that non-imaging clinical data alone permitted a satisfactory prediction of mortality (ROC-AUC: 0.87), but in the explainability analysis of our combined model, imaging features contributed to approximately 9% of the prediction. Several prior studies also similarly reported improvements in performance when adding chest radiographic data to clinical prediction models [25,26], but with lower ROC-AUC values for predicting mortality or disease progression with their combined imaging and clinical models (0.73–0.83) than those of our mortality prediction model. When considering only non-imaging clinical data, our ROC-AUC performance metric of 0.87 for in-hospital COVID-19 mortality is identical to the value of 0.87 reported using the A-DROP clinical criteria in a different dataset [27] and mildly higher than several other deep learning models (ROC-AUC: 0.84) [28] or clinical scoring methods to quantify pneumonia severity, such as qSOFA (ROC-AUC: 0.73) and CRB-65 (ROC-AUC: 0.80) [27]; that said, machine learning models with higher ROC-AUC values have been reported [29,30].

Among prior studies that quantified the relative contributions of mortality predictors when applying machine learning approaches to clinical data in COVID-19 is one study based on data at the time of ICU admission that found that the most relevant predictors of mortality were elevated BUN, Cr, INR, and RDW (red blood cell size variability); low albumin; gender; and age [31]. Our model differed in that we did not emphasize laboratory data as predictors and included both ICU and non-ICU inpatient encounters, but we similarly found that gender and age were among the top five contributing predictors. A meta-analysis of machine learning publications predicting disease severity in COVID-19 patients found that hypertension and diabetes were the most prevalent comorbidities among COVID-19 patients [32]. In our model, HbA1c, a marker of diabetes control, was the largest contributor to mortality risk. Our findings are corroborated by a recent meta-analysis showing a linear relationship between higher HbA1c values and increased mortality or disease worsening in COVID-19 [33].

A limitation of the study is that, due to the rapidly evolving and dynamic nature of the ongoing pandemic, prediction models developed with earlier clinical datasets likely have limited applicability to more recent COVID-19 cases, as it would not be possible to generalize to more recent variants or post-vaccination settings. Nonetheless, our described methodology and analyses represent reasonable approaches to quantify the contributions of multi-data inputs to an outcome prediction model that could be applied to updated clinical datasets for additional fine-tuning. The absence of recorded values for some variables in our dataset can be a source of bias that is challenging to address; in our study, restricting our analysis to only patients without missing data was not feasible due to the prevalence of missing data in our dataset population. A limitation of our analysis comparing relative contributions of different variables in our prediction model is that some variables could have had higher contributions to prediction if fewer entries had missing data. In addition, there may be potential for selection bias related to our use of only patients with chest radiographs. Moreover, as the explainability component of our current analysis did not include imaging features, we were not able to identify specific imaging features or areas of interest pertinent to the model's predictive performance. Other limitations of the study include the use of single-institution data and the use of encounter-level mortality that does not take into consideration the timing of chest radiographs relative to death or discharge. Future studies may address these limitations by offering additional analyses of transferability to datasets from other institutions and/or to encounters from later COVID waves.

**Author Contributions:** Conceptualization, X.V.N. and L.M.P.; methodology, X.V.N., E.D., S.C. and L.M.P.; software, E.D. and S.C.; validation, X.V.N., E.D., S.C. and L.M.P.; formal analysis, X.V.N., E.D., S.C., R.L.B. and L.M.P.; investigation, X.V.N., E.D., S.C. and L.M.P.; resources, L.M.P.; data curation, X.V.N., E.D. and S.C.; writing—original draft preparation, X.V.N., E.D. and S.C.; writing—review

and editing, X.V.N., E.D., S.C., R.L.B. and L.M.P.; visualization, X.V.N., E.D. and S.C.; supervision, X.V.N. and L.M.P.; project administration, X.V.N. and L.M.P.; funding acquisition, X.V.N. and L.M.P. All authors have read and agreed to the published version of the manuscript.

**Funding:** This research was funded by NIH/NIBIB, grant number 75N92020D00021.

**Institutional Review Board Statement:** Ethical review and approval were waived for this study due to exemption based on secondary use of a de-identified dataset.

**Informed Consent Statement:** Not applicable.

**Data Availability Statement:** Source data used in this study are available in a publicly accessible repository at https://doi.org/10.7937/TCIA.BBAG-2923. Accessed on 15 September 2021.

**Conflicts of Interest:** The funders had no role in the design of the study; in the collection, analyses, or interpretation of data; in the writing of the manuscript; or in the decision to publish the results. X.V.N. reports equity ownership in and dividends from the following publicly traded companies that may be considered "broadly relevant" to artificial intelligence: Alphabet, Amazon, AMD, Apple, Microsoft, and Nvidia. The authors declare no additional conflict of interest.

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
