# Peer review of "Mortality Prediction Analysis among COVID-19 Inpatients Using Clinical Variables and Deep Learning Chest Radiography Imaging Features"

_tomography, doi:10.3390/tomography8040151_

Round 1

Reviewer 1 Report

The authors, in this manuscript, attempt to develop a machine-learning model to quantify the relative contribution made by image- vs. non-image data to predict COVID19 outcomes (primary outcome as mortality); primarily focused on the pre-vaccine era.

Public, de-identified data were used to impute clinical characteristics. Then the data set was further trimmed based on the availability of chest radiographs.

1. There are patients without a sex assigned?  Prob should be dropped as it is missing a basic demographics (16% of inpatient death). This problem persists with all parameters in which 10+% of subjects have certain items undetermined ("not recorded").  This makes it difficult to assess how many patients actually have all entries in their information complete, and makes it unpredictable how the model will behave in the context of incomplete data.

2. Significant difference in the age distribution (inverse relationship) is noted.

3. While the addition of imaging may have "smoothen" the ROC, seems that AUC is basically the same.  This most likely occurred because the clinical data points imputed for the model development are very well established as risk factors for Cars-Cov2-related mortality.

4. Authors suggest that HA1c is most predictive of mortality outcomes, but with consistently missing data 10+% in other comorbidities, it is unclear how these "undefined" data points will push the model to choose DM as the most significant factor.  This may be an incorrect (premature) conclusion.  I would extensively discuss this pitfall and remove this statement.

5. Again higher mortality among ICU patients (if a patient requires ICU level care) is a well-known fact, that the model is just reconfirming.  This is based on selection bias.  The ICU group likely has all the "well-known" risk factors enriched in their demographics, so if the same factors are imputed for machine learning, it is anticipated that the ROC will improve (which is shown in Fig 5).  As a result, novelty on this is low.

6. Table 2 needs statistics comparing columns.

7.  During the early days of the pandemic, the use of CXR was limited due to difficulty isolating/quarantining these patients, so selecting those with available CXR selects likely those with a higher chance of mortality, which will likely enrich the subject population.  Unfortunately, this reduces the potential role of imaging during the early stages during which a machine-learning algorithm can identify those at risk of dying? Prior to "requiring CXR" to further evaluate "likely declining clinical conditions."

8. Many factors contributing to mortality are well-known factors, and while they are touted as risks to predict Sars-Cov2 death, they are also risks for mortality from other respiratory/ICU death.  Would have been better if there are non-COVID data to provide comparison and as controls.

9. I am not 100% sure why the authors censored the date during the non-vaccinated period.  Wouldn't the vaccination status be one of the most important factors (from 2021 and forward) when a model is developed?  Testing the model simultaneously during the vaccine period will provide if the model is simply reconfirming easily detectable risk factors for mortality (clinicians can figure this out within a few minutes of medical history) vs. actually informing clinicians with triage potentials that is critical to identify patients at risk for decision making.

After all, the importance of imaging seems mild at the most in this manuscript.

Author Response

Author responses

We thank the reviewers for their insightful comments on our manuscript.  We have made revisions to the manuscript, and our responses are provided below in bold text.

Reviewer 1

The authors, in this manuscript, attempt to develop a machine-learning model to quantify the relative contribution made by image- vs. non-image data to predict COVID19 outcomes (primary outcome as mortality); primarily focused on the pre-vaccine era.

Public, de-identified data were used to impute clinical characteristics. Then the data set was further trimmed based on the availability of chest radiographs.

  1. There are patients without a sex assigned? Prob should be dropped as it is missing a basic demographics (16% of inpatient death). This problem persists with all parameters in which 10+% of subjects have certain items undetermined ("not recorded"). This makes it difficult to assess how many patients actually have all entries in their information complete, and makes it unpredictable how the model will behave in the context of incomplete data.

In our analyses, categorical variables with missing values have missingness status encoded as such.  We therefore know exactly how many entries have values missing for each variable.  For predictive modeling purposes, it is often helpful to include missingness status as its own categorical level, especially if missingness status is overrepresented or underrepresented in certain patient populations and/or there are specific reasons why values may be missing.  We believe that model robustness is increased by including missingness status explicitly rather than ignoring it, as we are separating entries with known values from those with missing values in the model.

  1. Significant difference in the age distribution (inverse relationship) is noted.

Yes, not unexpectedly, older individuals are over-represented among patients showing in-hospital mortality.  The age contribution to mortality prediction in the final prediction model is discussed in Results.  In response to Item 6, we updated Table 2 to quantify statistical significance of the effects of these variables on in-hospital mortality.

  1. While the addition of imaging may have "smoothen" the ROC, seems that AUC is basically the same. This most likely occurred because the clinical data points imputed for the model development are very well established as risk factors for Cars-Cov2-related mortality.

We agree that some of the clinical variables, such as age, are known risk factors for COVID-related mortality.  However, AUC-ROC is slightly higher when including imaging data, and in our explainability analysis, the mean contributions of imaging to prediction are similar to age, gender, and BMI.

  1. Authors suggest that HA1c is most predictive of mortality outcomes, but with consistently missing data 10+% in other comorbidities, it is unclear how these "undefined" data points will push the model to choose DM as the most significant factor. This may be an incorrect (premature) conclusion. I would extensively discuss this pitfall and remove this statement.

Although we have modeled missingness status explicitly as a separate level of each categorical variable, it is possible that some variables could have had a higher contribution to prediction if fewer entries had missing data.  This has been added to the limitations section of the Discussion.

  1. Again higher mortality among ICU patients (if a patient requires ICU level care) is a well-known fact, that the model is just reconfirming. This is based on selection bias. The ICU group likely has all the "well-known" risk factors enriched in their demographics, so if the same factors are imputed for machine learning, it is anticipated that the ROC will improve (which is shown in Fig 5).  As a result, novelty on this is low.

While it is true that the ICU group has “well-known” risk factors that may favor mortality, this fact alone would not explain why mortality prediction maintains good discriminatory performance when applied to the ICU group.  We found that even within this high-risk pool of patients, the model can predict which of these high-risk patients survive.

  1. Table 2 needs statistics comparing columns.

A column has been added to Table 2 showing the chi square p-values for each of the categorical variables, with additional text added to Methods to describe the relevant statistical methods.

  1. During the early days of the pandemic, the use of CXR was limited due to difficulty isolating/quarantining these patients, so selecting those with available CXR selects likely those with a higher chance of mortality, which will likely enrich the subject population. Unfortunately, this reduces the potential role of imaging during the early stages during which a machine-learning algorithm can identify those at risk of dying? Prior to "requiring CXR" to further evaluate "likely declining clinical conditions."

We believe any selection bias resulting from our use of only patients with chest radiographs would be relatively minor.  Mortality for our subset of inpatients with chest radiographs was 21%, only slightly higher than for the entire dataset’s inpatient population (18%).  Moreover, we mitigated any data imbalance by training and testing on a population with equal numbers of fatal and nonfatal cases.  Nonetheless, we added a sentence to the limitations paragraph of Discussion.

  1. Many factors contributing to mortality are well-known factors, and while they are touted as risks to predict Sars-Cov2 death, they are also risks for mortality from other respiratory/ICU death. Would have been better if there are non-COVID data to provide comparison and as controls.

Our study is designed to predict all-cause mortality among COVID-positive individuals rather than COVID-specific mortality.  While it would be interesting to examine non-COVID data, the dataset we are using only contains COVID-positive individuals, concordant with our aims to assess prognosis among COVID-positive inpatients.

  1. I am not 100% sure why the authors censored the date during the non-vaccinated period. Wouldn't the vaccination status be one of the most important factors (from 2021 and forward) when a model is developed? Testing the model simultaneously during the vaccine period will provide if the model is simply reconfirming easily detectable risk factors for mortality (clinicians can figure this out within a few minutes of medical history) vs. actually informing clinicians with triage potentials that is critical to identify patients at risk for decision making.

We agree that vaccination status is important in predicting prognosis, but it is also helpful to assess mortality risk among unvaccinated individuals, especially since large pockets of the population remain unvaccinated.  Determining the effect of vaccination on mortality was not an aim of this study and was not possible with the dataset used in this study, which predominantly included encounters prior to widespread vaccine availability.  The resulting limitations related to applicability to more recent variants or for vaccinated individuals have been discussed in the Discussion section.

After all, the importance of imaging seems mild at the most in this manuscript.

We agree that the role of imaging is mild, contributing to only 9% of prediction.  However, to our knowledge, the predictive role of imaging relative to clinical variables within this context has not been quantified in this manner, and we believe our manuscript offers new insights for the journal’s readership.

Reviewer 2 Report

This paper is well written and has sound scientific merits. I would like to encourage the authors to add sensitivity and specificity in the result section.

Author Response

Author responses

We thank the reviewers for their insightful comments on our manuscript.  We have made revisions to the manuscript, and our responses are provided below in bold text.

Reviewer 2

This paper is well written and has sound scientific merits. I would like to encourage the authors to add sensitivity and specificity in the result section.

                Sensitivity and specificity values have been added to results.

Reviewer 3 Report

The authors developed an AI algorithm to predict inpatient mortality among COVID-19 patients using clinical and chest radiographic data. Non-imaging predictors included demographics, pre-admission clinical history, and past medical history variables. Imaging features were extracted from chest radiographs by applying a deep convolutional neural network with transfer learning. A multi-layer perceptron combining deep learning features from chest radiography with patient clinical features was trained to predict mortality. The Local Interpretable Model-Agnostic Explanations (LIME) method was used to explain model predictions.

The work is well written and very clearly presented.

Only pagination errors (pages 4, 6, 7) can be indicated.

Author Response

Author responses

We thank the reviewers for their insightful comments on our manuscript.  We have made revisions to the manuscript, and our responses are provided below in bold text.

Reviewer 3

The authors developed an AI algorithm to predict inpatient mortality among COVID-19 patients using clinical and chest radiographic data. Non-imaging predictors included demographics, pre-admission clinical history, and past medical history variables. Imaging features were extracted from chest radiographs by applying a deep convolutional neural network with transfer learning. A multi-layer perceptron combining deep learning features from chest radiography with patient clinical features was trained to predict mortality. The Local Interpretable Model-Agnostic Explanations (LIME) method was used to explain model predictions.

The work is well written and very clearly presented.

Only pagination errors (pages 4, 6, 7) can be indicated.

                Thank you for your feedback.

Round 2

Reviewer 1 Report

I don't agree with the author's rebuttal that the model can adjust for missing data because the direction of the missing data can be two opposite directions, and I can't see how computer modeling can predict that direction no matter how good it is.  All I was asking was to remove incomplete data so that the data set can be cleaner to rid of bias in model development.

It seems that based on the authors' responses, the current manuscript version may be the best version while keeping all available data.  

Author Response

Thank you for raising this point, but unfortunately, excluding patients with incomplete data would not have been feasible for our dataset, since all patients had at least one missing entry among the clinical variables examined.  Furthermore, depending on the randomness in distribution of missing entries, removing participants with missing data might not necessarily produce unbiased estimates.  We do acknowledge that our approach does not eliminate bias related to presence of missing data and have added additional verbiage to the Limitations section to further discuss this bias and the infeasibility of performing a complete-case analysis.  Because it is often important for decision systems to be able to operate in the context of missing clinical data, we felt that designating missingness status as a separate level of each categorical variable was a reasonable approach.  We added text to Methods describing our justification for handling missing data in this fashion.